# A reliable *in vitro* rumen culture system and workflow for screening anti-methanogenic compounds

Philip P. Laric[2], Armina Mortazavi[2,3], Ewa Węgrzyn[4], Kathrin Simon[2], Pauline S. Rittel[2,5], Florian M. Trefz[5], Benedikt Sabass [1,2,3]*

**1** Department of Physics, TU Dortmund University, Dortmund, Germany, **2** Department of Veterinary Science, LMU Munich, Germany, **3** Faculty of Physics and Center for NanoScience, LMU Munich, Germany, **4** Institute for Chemical Epigenetics (ICEM), LMU Munich, Germany, **5** Clinic for Ruminants with Outpatient Clinic and Herd Management, Center for Clinical Veterinary Medicine, LMU Munich, Oberschleißheim, Germany

* benedikt.sabass@tu-dortmund.de

## Abstract

Arguably the biggest man-made challenge of the century is halting climate change. Livestock's methane ($CH_4$) emissions, a greenhouse gas with a higher global warming potential than carbon dioxide ($CO_2$), represent a prime target for reducing anthropogenic impact. While the reduction of enteric methane emissions through feed additives has been demonstrated, potent and affordable compounds inhibiting methanogenesis in ruminants are not yet well established. Reliable methods for reproducible cultivation of the rumen microbiome in the laboratory are an essential tool for the study of methanogenesis. We have developed a versatile setup that allows for the cultivation of the ruminal microbiome in a benchtop configuration and combines, miniaturizes, and improves existing systems. The design is based on standard laboratory equipment, including bottles, serological pipettes, tubing, and Luer-Lock valves. The apparatus enables long-term cultivation of primary cultures extracted from the rumen of slaughtered cattle. We describe rumen content acquisition, preparation, the cultivation procedure, and demonstrate the system's performance. The efficacy of the system is demonstrated through the administration of various concentrations of state-of-the-art methanogenesis inhibitors. These inhibitors include lyophilized *Asparagopsis taxiformis* (AT), bromoform (BF), iodoform (IF), 3-nitrooxypropanol (3-NOP), rapeseed oil, and BF dissolved in rapeseed oil, with maximum $CH_4$ reductions of 96.29% *(p = 5.00E-05, Cohen's d = 30.29)*, 98.22% *(p = 2.88E-05, d = 23.07)*, 96.26% *(p = 1.03E-05, d = 30.29)*, 74.63% *(p = 8.88E-05, d = 13.32)*, 28.96% *(p = 0.001, d = 3.99)*, and 98.51% *(p = 4.18E-06, d = 39.94)*, respectively, in comparison to the negative control. The gas production dynamics in our setup align with previously published results, which supports the validity of the system. Compared to conventional methodologies, the described setup offers enhanced versatility and ease of

**Data availability statement:** All sequencing files are available from the NCBI SRA database (accession number PRJNA1255180).

**Funding:** This Project was funded by the German Federal Ministry of Education and Research (BMBF) under grant number 031B1504. PL and BS received funding from the European Union's Horizon 2020 research and innovation programme (grant agreement No 852585). The funders had no role in study design, data collection and analysis, decision to publish, or preparation of the manuscript.

**Competing interests:** We have read the journal's policy, and the authors of this manuscript have the following competing interests: The authors PL and BS are inventors on patent [AJ2811 PCT, Methods and compositions for reducing gas emission, and/or methane emission, and/or improving feed utilization of ruminants, European Patent Office, The Hague] related to the reduction of methane emission in ruminants. This does not alter our adherence to PLOS ONE policies on sharing data and materials.

**List of abbreviations:** 16S rDNA, 16S ribosomal DNA; 3-NOP, 3-Nitrooxypropanol; AT, *Asparagopsis taxiformis*; BF, Bromoform; BFOIL, Bromoform in rapeseed oil; $CH_4$, Methane; $CO_2$, Carbon dioxide; $CO_2e$, Carbon dioxide equivalent; d, Cohen's d; $\delta$, Cliff's delta; DNP, 2,2-dimethyl-3-(nitrooxy) propanoic acid; GHG, Greenhouse gas; GWP, Global warming potential; gDM, Grams dry matter; $H_2$, Hydrogen; IF, Iodoform; MCR, Methylcoenzyme M reductase; NAD, Nicotinamide adenine dinucleotide; NC, Negative control; NCOIL, Negative control rapeseed oil; NPD, N-[2-(nitrooxy)ethyl]-3-pyridinecarboxamide; OM, Organic matter; p, p-value; PCR, Polymerase chain reaction; RUSITEC, Rumen simulation technique; SEM, Standard error of the mean.

use. Furthermore, Fourier-transform-infrared-spectroscopy is implemented in a novel and low-cost approach for quantifying $CH_4$ and $CO_2$ in the headspace gas. Together, these methodological advances provide an accessible and reproducible platform for long-term *in vitro* rumen cultivation for the screening of anti-methanogenic additives.

## Introduction

The goal of achieving net-zero greenhouse gas emissions by the second half of this century was established in the Paris Agreement of 2015 [1]. Methane ($CH_4$) plays a particularly important role in achieving this goal [2]. The atmospheric $CH_4$ concentration has tripled since the onset of the industrial revolution in the late 1700s [3]. Methane has a relatively short atmospheric half-life of $9.1 \pm 0.9$ years [4], but a much higher short-term global warming potential (GWP) when compared to carbon dioxide ($CO_2$). It is estimated that 1 gram of $CH_4$ has the GWP of 34 grams of $CO_2$ on a time horizon of 100 years, but 86 grams of $CO_2$ over 20 years [5]. Consequently, the reduction of $CH_4$ emissions provides a promising avenue to mitigate global warming.

Livestock production is responsible for approximately 6.38% of global greenhouse gas (GHG) emissions, equivalent to 3.1 Gt of $CO_2e$ ($CO_2$ equivalents) per year, mostly in the form of $CH_4$ and nitrous oxide. This includes enteric fermentation, manure, and rice feed [6–8]. With global livestock production projected to continue to rise, the reduction of these emissions is of increasing urgency. The digestive system of ruminants contains a complex microbiome consisting of anaerobic bacteria, archaea, protozoa, and fungi. These organisms enable their host to extract nutrients from otherwise mostly indigestible parts of plants. In this symbiosis, the rumen microbiota degrade substrates rich in polymerized structural carbohydrates of otherwise low nutritional value [9]. The main fermentation end products are short-chain fatty acids such as propionate, butyrate, acetate, and the gases $CO_2$ and hydrogen ($H_2$). Elevated concentrations of $H_2$ inhibit, for example, the redox cycling of nicotinamide adenine dinucleotide (NAD), so $H_2$ must be effectively removed from the system. The most important $H_2$ sink is its use as a substrate in the metabolism of methanogens in the rumen. Methanogens obtain energy for their growth by the reduction of various substrates, such as $CO_2$, to $CH_4$ with the aid of $H_2$ [10]. Suppression of methanogenesis has been shown to result in a redirection of metabolic $H_2$ pathways [11,12]. For instance, *Ruminococcus albus*, one of the major plant cell-wall degrading bacteria, responds to an increase in the partial pressure of $H_2$ by a change in the pattern of fermentation products. In the absence of methanogenic archaea, less $H_2$ is produced. The normally dominant metabolism of cellulose to butyrate or acetate is then replaced by metabolic pathways that produce less $H_2$, such as propionate or valerate [13]. As a result of the eructation of $CH_4$, ruminants lose approximately 6% of their energy intake [14]. Digestion parameters associated with different levels of $CH_4$ production depend on a complex interplay between the microbiome, the animal, and the feed. Feed processing techniques, used at the farm level, as well as the specific nutrients present in feed formulations, significantly affect $CH_4$ production [14]. In

addition, genetic variation among animals contributes to differences in $CH_4$ emissions [15], underscoring the complexity of methanogenesis in ruminants, and the necessity of *in vitro* systems for clarifying the mode of action of different feed additives.

There is a wide range of feed additives available to manipulate the rumen microbiome including phytochemicals and essential oils, red algae, nitrates, ionophores, haloalkanes, halogenated sulfonate compounds, nitrooxy compounds, bacteriocins, addition of acetogens, and methods to control the protozoan population [16]. To provide context for this work, we briefly review a few prominent anti-methanogenic additives.

One extensively studied group of inhibitors of microbial methanogenesis are polyhalogenated compounds that can be characterized as $CH_4$ analogs. Initial studies investigated the use of carbon tetrachloride and chloroform [17]. Means to reduce methanogenesis were proposed to protect sheep from the harmful effects of *Heliotropium europaeum*, a plant that contains hepatotoxic pyrrolizidine alkaloids, whose degradation is favored by elevated $H_2$ levels [18–20]. Further polyhalogenated, anti-methanogenic compounds include chloral hydrate [21], tribromoacetaldehyde, bromoform (BF), and carbon tetrabromide [22–25]. A recent study with iodoform (IF) showed good anti-methanogenic activity, but as a side effect, milk yield and feed intake were reduced. Furthermore, alterations of several metabolic markers were indicative of the presence of a negative energy balance. Reduced feed intake may result directly from the administered IF, an excess of iodine, as iodine was also supplemented, or a refusal of the diet due to palatability issues [26].

Promising natural anti-methanogenic feed additives are red algae, in particular *Asparagopsis taxiformis* (AT). The main active agent in AT is bromoform. Supplementation of AT can result in up to 80% reduction of $CH_4$ production, and a reduction in feed intake with no apparent negative effect on average daily weight gain [27,28]. However, high concentrations of BF are considered toxic. There are legitimate concerns about environmental side effects, including potential impacts on the ozone layer, because large-scale production and feeding of AT could increase halocarbon emissions [29]. Furthermore, supplementation of BF containing compounds can lead to the formation of volatile dibromomethane and bromomethane in the rumen [30]. Moreover, AT supplementation has been shown to significantly increase bromine and iodine levels in milk, approaching or exceeding recommended limits. Such findings highlight the importance of environmental and food-safety considerations [31].

An entirely different group of anti-methanogenic compounds are nitro esters. The well-studied small molecule 3-nitrooxypropanol (3-NOP) inhibits Methyl-Coenzyme-M-Reductase (MCR). This enzyme catalyzes the final step in the formation of $CH_4$ by combining the $H_2$ donor coenzyme B and the methyl donor coenzyme M, see Fig 1 [32]. Suppression of methanogenesis is thought to occur through specific binding of 3-NOP to the active center of MCR, where it oxidizes Ni(I) [33]. Other recently discovered anti-methanogenic compounds containing nitro esters include N-[2-(nitrooxy)ethyl]-3-pyridinecarboxamide (NPD), 2,2-dimethyl-3-(nitrooxy)propanoic acid (DNP), and nitroglycerin. Notably, all of these substances show distinct inhibitory properties due to their molecular structures, with nitroglycerin, which contains three nitrooxy moieties, exhibiting the strongest inhibitory activity [34].

There is a considerable amount of published research on ruminal fermentation, which typically focuses on evaluating feed value and digestion efficiency. *In vitro* experiments are frequently done with cultures from rumen inocula, see Fig 2. Rumen microbiome cultivation methods can be categorized into two different approaches, namely, short-term batch cultivation and long-term rumen simulation techniques (RUSITEC), both of which will be discussed below. The batch culture system is characterized by its ease of use with possible high sample throughput. Batch culture systems have been designed to evaluate the short-term effects of a compound on fermentation parameters [17,19]. The integration of an automated gas accumulation recording and release system (ANKOM) has further expanded the utility of laboratory flasks in batch experiments [27]. For operation, rumen inoculum is diluted with buffer and transferred to vessels prepared with different compounds of interest and the substrate. The vessels are subsequently incubated at 39°C. Typically, the duration of these batch experiments is up to three days, until the culture has fully depleted the nutrients from the substrate. Recently, automated high-throughput systems in well plate format have become available [35,36], with the potential to miniaturize the

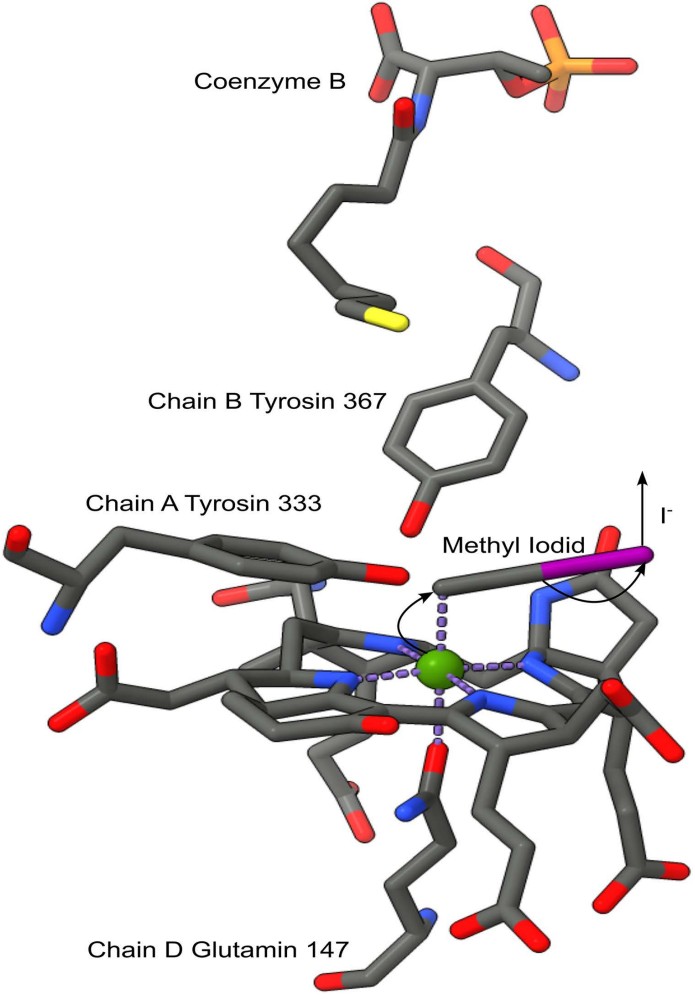

Coenzyme B

Chain B Tyrosin 367

Chain A Tyrosin 333

Methyl Iodid

I⁻

Chain D Glutamin 147

**Fig 1. X.ray-crystallography model of the active center of the MCR; cofactor F430 with methyl iodide and Coenzyme B. Grey: carbon; blue: nitrogen; red: oxygen; yellow: sulfur; pink: iodide; green: nickel(III) [69].**

batch setup to a 96-well plate format, thereby enhancing the throughput. However, miniaturization imposes constraints on the parameters that can be analyzed, confining them to gas production parameters and the nutrients supplied. The fundamental limitation of the batch culture system is its incapacity to sustain a stable microbiome, thus preventing long-term experiments to assess the effects on the microbiome and the consistency of the treatment effects. The decline in certain species observed during the experiments is indicative of a diminished quality of the measurements [37].

RUSITEC setups are designed to simulate the conditions of the rumen in more detail. They consist of a series of multiple vessels that mimic the rumen environment and maintain a more stable microbiome that can closely resemble the original rumen inoculum [38]. Despite the enhanced stability of the microbiome in rumen simulation fermenters, a significant loss of protozoal biomass has been documented [39]. In recent years, various systems capable of sustaining the whole microbiome during prolonged cultivation have been designed. Examples of such state-of-the-art systems include semi-continuous single-flow fermenters [40], continuous dual-flow fermenters that retain some of the solids effluent by filtration [41] and intermittent single-flow systems that stratify the fermenter contents by slow or intermittent agitation [42]. The semi-continuous single-flow fermenter represents the most prominent approach. In this configuration, the diet is stored in a permeable bag within the system, and a

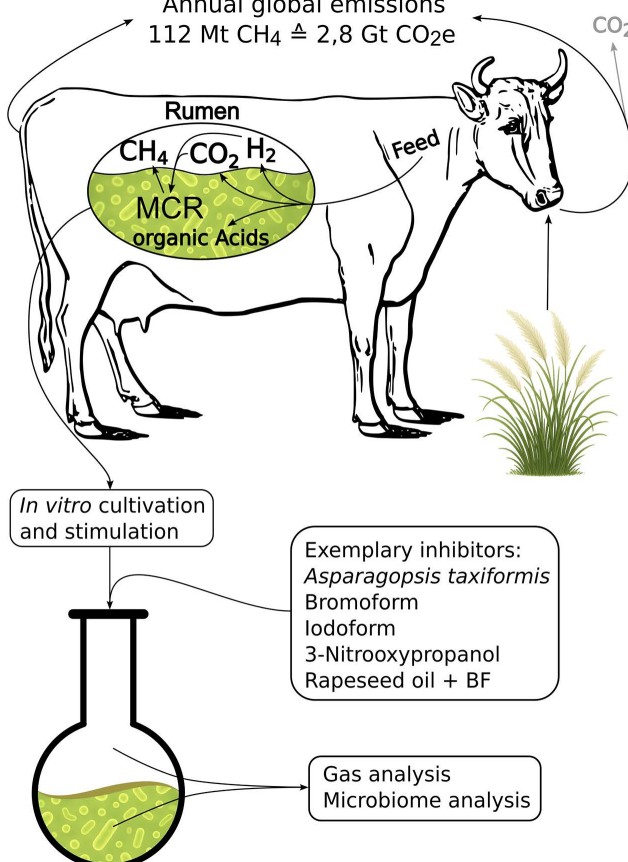

**Fig 2. Methanogenesis in the cattle rumen and its annual methane emissions.** Overview of the workflow for inhibitor development *in vitro*.

system of tubes and pumps facilitates the constant exchange of culture and fresh buffer. Regular feeding is achieved by open-ing the system and the replacement of the bag inside [40]. From a practical point of view, a limitation of the RUSITEC approach is the low sample throughput, because the complicated and expensive systems cannot be easily multiplexed.

We report here the design of a reliable and easy-to-use, semi-continuous rumen simulation system that is optimized for testing anti-methanogenic supplements. Detailed assembly instructions are provided, and the setup is designed to be built at minimal cost. To achieve this, we used common laboratory equipment that was crafted into an anaerobic incubation apparatus. Batch cultivation approaches were combined with RUSITEC approaches to create a continuous batch system. The integrated inlets and outlets facilitate daily probing and feeding without disturbing the system, as well as stimulation with compounds of interest. To validate the setup, we compared $CH_4$, $CO_2$, and residual-gas production in response to differ-ent concentrations of well-studied anti-methanogenic supplements. In detail, we compared AT [27], BF [43–45], IF [22,26], 3-NOP [46–49], rapeseed oil, and BF dissolved in rapeseed oil [50,51], while tracking shifts in the microbiome. This work therefore provides a much-needed direct comparison of the most prominent inhibitors [52,53] and delivery formulations [54].

## Materials & methods

The protocol described in this peer-reviewed article is published on protocols.io, https://dx.doi.org/10.17504/protocols.io.kxygxwrnkv8j/v1 and is included for printing as supporting information as S1–S6 File with this article.

Data were evaluated using two-sided t-tests for parametric data and Mann–Whitney U-tests for nonparametric data. Where applicable, the standard error of the mean (SEM) was calculated as a measure of variance. Effect sizes were quantified as Cohen's $d$ for parametric datasets and as Cliff's delta (δ) for non-parametric comparisons. Cohen's $d$ expresses the standardized mean difference between groups, while Cliff's delta quantifies ordinal dominance.

## Anticipated results

To facilitate basic research on methanogenesis in ruminants and to provide a tool for the development of state-of-the-art anti-methanogenic compounds, we designed an incubation setup for daily anaerobic feeding and sampling. The setup also allows for the conduction of tests on gas production and microbiome sampling. To demonstrate the possibility to culti-vate anaerobic cultures for extended periods, weeks-long *in vitro* studies were performed. In this section, we demonstrate a comparison of the anti-methanogenic effects of the additives AT powder, BF, IF, 3-NOP, rapeseed oil ($NC_{OIL}$), as well as a BF solution in oil ($BF_{OIL}$). The investigation also encompassed an analysis of these agents' influence on various other parameters in comparison to a negative control (NC). For each run of replicate experiments, all compounds were tested on cultures from a single inoculum prepared for each run by mixing equal parts of rumen contents from four animals. Four separate runs of experiments were conducted to produce data replicates.

## Effect of additives on total gas production

We first checked whether anti-methanogenic compounds directly affect the total gas production. The cumulative gas production of the NC was 2187.25 mL *(SEM = 36.89 mL)* within 7 days of incubation. The produced gas volumes were summed for those days of the experiment after stimulation when the tested anti-methanogenic compounds resulted in changes in $CH_4$ production and were compared to the overall volumes of gas production of the NC on these days. The results demonstrated that 3 mg and 6 mg AT both resulted in a slight increase of 5.73% *(p > 0.025, d = 1.4)* and 5.19% *(p > 0.05, d = 0.75)* in gas production, respectively, whereas 9 mg AT resulted in a significant reduction of 17.05% *(p = 0.0006, d = 5.73)*. In contrast, stimulations with 2.5 µM and 3.75 µM BF yielded significant reductions of 15.38% *(p = 0.002, d = 3.7)* and 9.90% *(p = 0.004, d = 3.40)*, respectively. Similarly, 2.5 µM and 3.75 µM IF resulted in significant reductions of 12.46% *(p = 0.001, d = 4.12)* and 13.03% *(p = 0.002, d = 4.59)*, respectively. The 1.25 µM $BF_{OIL}$ treatment yielded a reduction of 8.64% *(p = 0.038, d = 1.96)* and 10.18% *(p = 0.033, d = 1.95)* in total gas production compared to the NC and the $NC_{OIL}$, respectively. No further significant changes in total gas production were observed.

## Effect of additives on methane production

To assess the anti-methanogenic activity, the mean of all daily produced $CH_4$ fractions was calculated for compound tests and for simultaneously conducted negative control experiments. The mean percentage of $CH_4$ in the gas produced by the negative control from day 3 to day 11 of the incubation period, was 20.06% *(SEM = 0.84%)*. The $CH_4$ production, with the standard error of the mean, normalized on the corresponding negative control, is shown in Fig 3. All compound supple-mentations yielded a significant reduction in the $CH_4$ production rate compared to the corresponding negative control. The $CH_4$ reduction varied depending on the compound and its respective concentration. Table 1 provides an overview of the results. Increasing concentrations of AT, BF, and IF produced greater and more sustained $CH_4$ inhibition. AT, BF, and IF treatments at the highest concentrations achieved maximum $CH_4$ reductions of 96.29% on day 5 *(p = 5.00E-06, d = 30.29)*, 98.22% on day 6 *(p = 2.88E-05, d = 23.07)*, and 96.26% on day 5 *(p = 1.03E-05, d = 30.82)*, respectively. All three main-tained significant inhibition for 7 days, with an average inhibition of 83.46% *(p = 4.34E-05, d = 18.79)*, 86.24% *(p = 2.11E-05, d = 20.01)*, and 81.90% *(p = 0.003, d = 17.00)*, respectively. At 1.25 µM, comparison indicated one significant difference between BF and IF *(p = 0.031)*. However, due to variability across replicates and overlapping standard errors, no consis-tent differences could be established. At higher concentrations (2.5 µM and 3.75 µM), BF and IF did not differ significantly *(p > 0.05)*. For 33 µM 3-NOP, $CH_4$ inhibition was moderate, with a maximum reduction of 74.63% on day 2 *(p = 8.88E-06,*

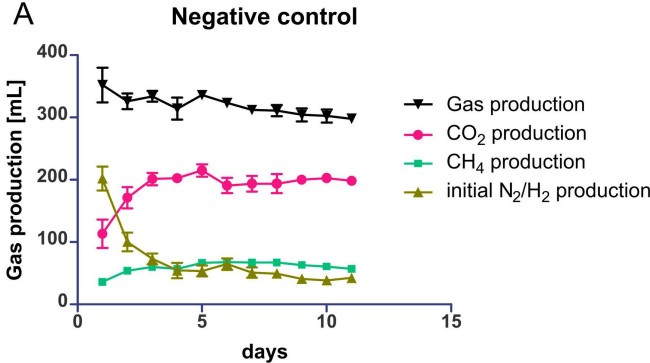

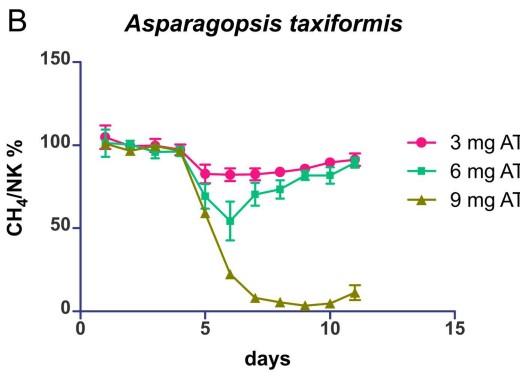

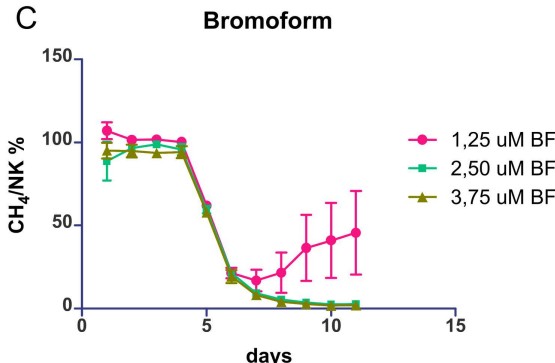

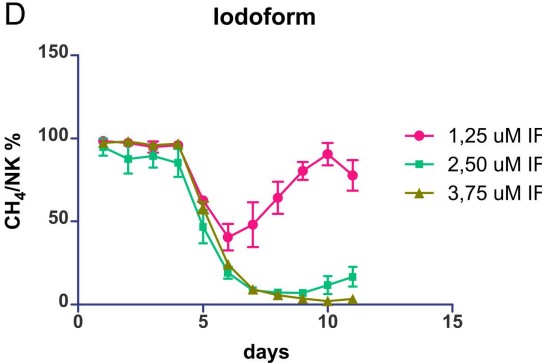

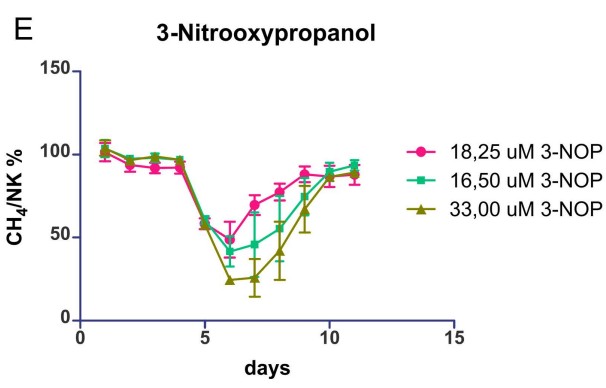

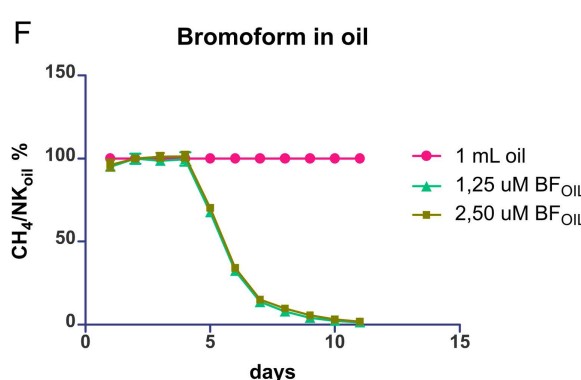

**Fig 3. Comparison of efficacy of different anti-methanogenic additives. (A)** Gas production in the negative control. Produced $CH_4$ volumes for the stimulation experiments (B-F) are normalized by $CH_4$ volumes produced in simultaneously conducted negative control experiments. Tested additives are AT: *Asparagopsis taxiformis*, BF: bromoform, $NC_{OIL}$: rapeseed oil, IF: iodoform, 3-NOP: 3-nitrooxypropanol, $BF_{OIL}$: bromoform in rapeseed oil.; **(B)-(F)**

The pink circles represent the lowest inhibitor concentration, the green squares the medium concentration and the brown triangles the highest inhibitor concentration. Error bars represent ± SEM; n = 4 experiment repetitions.

**Table 1. Effect of stimulation with different anti-methanogenic additives on the production of $CH_4$.** The reductions are normalized to the corresponding NC and given in percent of the NC. The column labelled "Duration of $CH_4$ reduction" represents the number of days following stimulation during which $CH_4$ reduction was reduced.

| Compound | Maximum $CH_4$ reduction | Duration of $CH_4$ reduction (days). Data recorded for 7 days after stimulation | Day of Maximum Inhibition |
|---|---|---|---|
| AT | | | |
| 3 mg | 17.67% (p = 0.004, d = 2.95) | 6 (p = 0.02, d = 2.17) | 2nd |
| 6 mg | 46.07% (p = 0.03, d = 2.71) | 7 (p = 0.02, d = 3.18) | 2nd |
| 9 mg | 96.29% (p = 5.00E-06, d = 30.29) | 7 (p = 4.34E-05, d = 18.79) | 5th |
| BF | 7 d | | |
| 1.25 µM | 83.24% (p = 1.45E-04, d = 8.74) | 6 (p = 0.009, d = 5.89) | 3rd |
| 2.5 µM | 97.39% (p = 3.01E-05, d = 22.89) | 7 (p = 0.004, d = 2.95) | 6th & 7th |
| 3.75 µM | 98.22% (p = 2.88E-05, d = 23.07) | 7 (p = 2.11E-05, d = 20.01) | 6th |
| IF | | | |
| 1.25 µM | 59.25% (p = 0.001, d = 4.72) | 5 (p = 0.009, d = 3.76) | 2nd |
| 2.5 µM | 92.85% (p = 4.57E-08, d = 23.24) | 7 (p = 8.67E-04, d = 14.97) | 4th & 5th |
| 3.75 µM | 96.26% (p = 1.03E-05, d = 30.82) | 7 (p = 0.003, d = 17.00) | 5th |
| 3-NOP | | | |
| 8.25 µM | 50.62% (p = 0.009, d = 2.91) | 4 (p = 0.008, d = 3.30) | 2nd |
| 16.5 µM | 58.03% (p = 0.002, d = 4.00) | 6 (p = 0.02, d = 2.76) | 2nd |
| 33 µM | 74.63% (p = 8.88E-06, d = 13.32) | 7 (p = 0.01, d = 5.07) | 2nd & 3rd |
| NC$_{OIL}$ | | | |
| 1 mL | 28.96% (p = 0.001, d = 3.99) | 7 (p = 0.008, d = 3.52) | 4th |
| BF$_{OIL}$ | | | |
| 1.25 µM | 98.51% (p = 4.18E-06, d = 39.94) | 7 (p = 3.47E-04, d = 17.54) | 7th |
| 2.5 µM | 98.02% (p = 5.20E-06, d = 39.97) | 7 (p = 3.22E-04, d = 16.97) | 7th |

d = 13.32), lasting up to 7 days (p = 0.16, d = 5.07). Lower concentrations of 3-NOP (8.25 µM and 16.5 µM) resulted in weaker effects, with $CH_4$ reductions of 50.62% on day 2 (p = 0.009, d = 2.91) and 58.02% on day 2 (p = 0.003, d = 4.00), respectively. The impact of rapeseed oil alone was found to be limited, with a 28.96% (p = 0.001, d = 3.99) reduction in $CH_4$ on day 8. However, BF dissolved in rapeseed oil suppressed $CH_4$ production effectively. Concentrations of 1.25 µM and 2.5 µM BF in oil achieved reductions of 98.51% (p = 4.18E-06, d = 39.94) and 98.02% (p = 5.20E-06, d = 39.97) on the last day of incubation, respectively, and persisted for the full 7-day incubation period (p = 0.0003, d = 17.54) and (p = 0.0003, d = 16.97), respectively. Table 1 also shows the day of maximum inhibition to quantify the time delay until the start of recovery of $CH_4$ after stimulation.

## Effect of additives on carbon dioxide production

Using Fourier-transform-infrared-spectroscopy, the averages of the daily produced $CO_2$ volumes in the headspace were calculated. The mean daily $CO_2$ production of the negative control, from day 3 to day 11 of the incubation period, was 63.52% (SEM = 2.51%). Slight, non-significant (p > 0.025), increases were recorded for 9 mg AT and 3.75 µM BF with 13.46% (d = 2.29) and 7.18% (d = 1.70), respectively. A significant decrease of 9.07% (p = 0.009, d = 2.68) in $CO_2$ production occurred for stimulation with 33 µM 3-NOP. In contrast, a significant increase in $CO_2$ production was observed for 2.5

μM BF, 3.75 μM IF, and 1 mL rapeseed oil, with values of 12.43% *(p = 0.002, d = 4.35)* 16.57% *(p = 0.004, d = 4.03),* and 8.80% *(p = 0.013, d = 2.10)*, respectively.

## Effect of additives on the residual gas composition

The produced volumes of residual gas were calculated daily along with their average concentration. At the start of cultivation, the bottle headspace was flushed with pure nitrogen to ensure anaerobic conditions. As a result, gas values measured on the first day reflect this artificial dilution, and only values recorded after stimulation (day 4) were considered for analysis. The average residual gas production of the NC was 15.59% *(SEM = 2.47 mL)* within 7 days of incubation. The stimulations with 9 mg AT, 1.25 μM BF, 2.5 μM BF, 3.75 μM BF, 2.5 μM IF, 3.75 μM IF, and 1.25 μM BF and 2.5 μM BF in rapeseed oil resulted in a significant increase in IR-silent molecular species, as shown in Table 2. We consider it likely that the residual gas is mainly $H_2$.

## Repeated stimulation over 23 days of cultivation

To provide an initial assessment of the reversibility of methanogenesis inhibition, a repeated stimulation experiment was conducted. For simplicity, we extended the duration of the fourth repetition of the experiments used to accumulate statistical data for this work. The cultivation period was extended to a total duration of 23 days. After a brief regeneration phase after the first stimulation, a second stimulation was performed on day 16 of the experiment. The result was an effect similar to the first stimulation on day 4. For almost all additives, $CH_4$ had recovered and decreased again after the second stimulation. On average, $CH_4$ inhibition was 6% lower than in the initial stimulation. Exceptions were $BF_{OIL}$ treated samples, which did not recover their $CH_4$ production after the initial stimulation (Data not shown; n = 1).

## Microbiome analysis

For an evaluation of the stability of the microbiome with and without the influence of anti-methanogenic additives, the relative microbial abundance was assessed via 16S rDNA metagenomics. Samples were taken prior to stimulation on days 2 and 4. Subsequently, on day 8 where the compounds showed activity, and on day 11, where $CH_4$ production was typically recovering. The most prevalent taxa found in our samples were Bacteroidota, Euryarchaeota, Fibrobacterota, Firmicutes, Proteobacteria, Spirochaetota, and Verrucomicrobia. The relative abundance of the major taxa is shown in Fig 4. Taxa that were present in smaller numbers were grouped as Other, comprising Actinobacteriota, Bdellovibrionota, Campilobacterota, Cyanobacteria, Chloroflexi, Desulfobacterota, Elusimicrobiota, Halobacterota, Patescibacteria, Planctomycetota,

Table 2. Effect of the stimulation with different anti-methanogenic additives on the production of residual gases. The increases are normalized to the corresponding NC and are presented as an increase compared to the NC. The duration indicates the number of days following stimulation during which $H_2$ production was increased.

| Compound | Maximum increase in residual gas (likely $H_2$) | Duration (days) and average of increased residual gas production |
|---|---|---|
| 9 mg AT | 69.58% *(p = 0.006, d = 3.09)* | 1 |
| 1.25 μM BF | 79.52% *(p = 0.02, d = 1.86)* | 2; 78.89% *(p = 0.02, d = 1.87)* |
| 2.5 μM BF | 94.78% *(p = 0.02, d = 1.94)* | 5; 85.50% *(p = 0.01, d = 2.61)* |
| 3.75 μM BF | 137.64% *(p = 0.005, d = 2.95)* | 6; 111.56% *(p = 0.008, d = 2.99)* |
| 2.5 μM IF | 134.16% *(p = 0.01, d = 2.33)* | 3; 122.31% *(p = 0.009, d = 2.57)* |
| 3.75 μM IF | 80.48% *(p = 0.004, d = 3.23)* | 3; 67.42% *(p = 0.009, d = 3.31)* |
| 1.25 μM BF Oil | 112.55% *(p = 5.15E-4, d = 4.23)* | 3; 100.75% *(p = 0.003, d = 3.35)* |
| 2.5 μM BF Oil | 87.07% *(p = 0.007, d = 2.59)* | 3; 87.40% *(p = 0.004, d = 2.98)* |

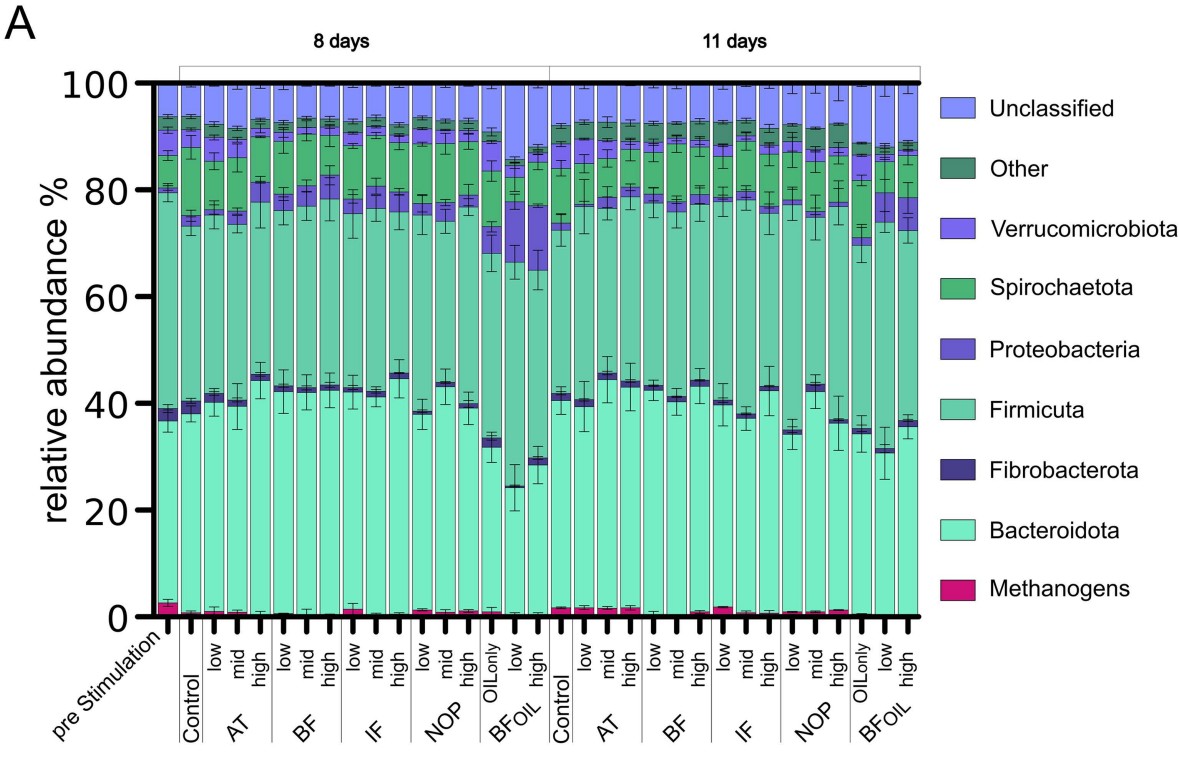

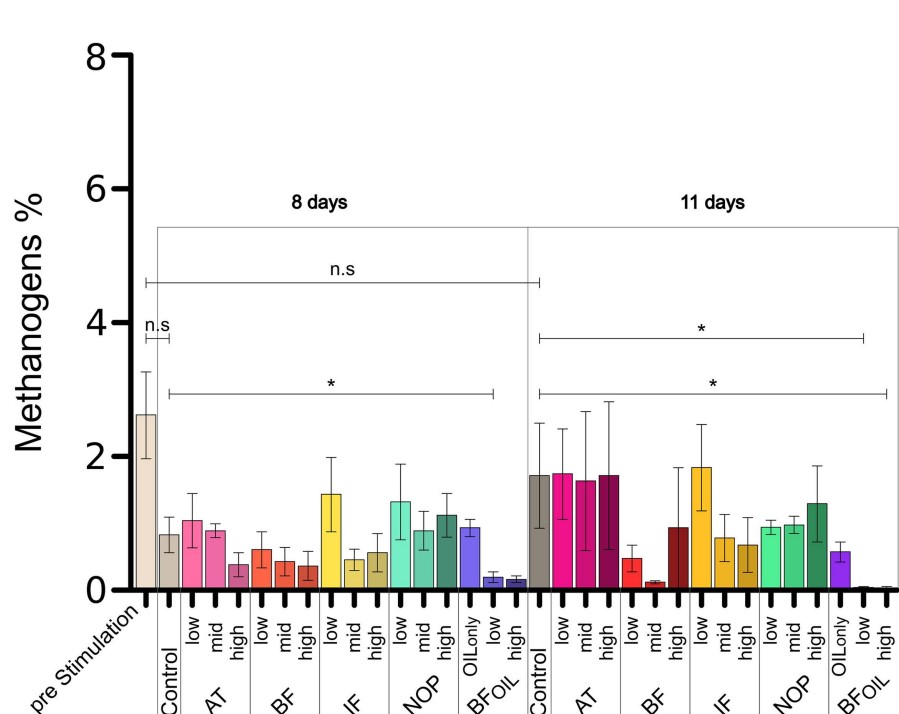

**Fig 4. Comparison of relative abundance of microbial taxa. (A)** Relative abundance of microbial taxa at the phylum level. **(B)** Relative abundance within the Methanogens/the Euryarchaeota phylum. BF: bromoform, IF: iodoform, 3-NOP: 3-nitrooxypropanol, $BF_{OIL}$: bromoform in rapeseed oil. Error bars represent ± SEM; n = 4 experiment repetition. Asterisk "*" indicates $p < 0.05$.

Synergistota, and WPS-2. All unclassified sequence counts were combined into a single category, designated as "Unclassified". The complete list of reported taxa can be found in the supplementary information (S2 Table).

Over the course of the experiment, the taxa of Bacteroidota and Firmicutes exhibited a stable abundance with a mean of 36.32% *(SEM = 2.29%)* and 35.61% *(SEM = 3.31%)*, respectively. Verrucomicrobia, Other, and Unclassified had mean abundances of 4.10% *(SEM = 0.47%)*, 2.81% *(SEM = 0.42%)*, and 6.42% *(SEM = 0.73%)* respectively. Fibrobacterota showed a large variability in the inoculum, with a mean on day 2 of 4.45% *(SEM = 0.73%)*. The overall abundance for Fibrobacterota was 2.66% *(SEM = 1.49%)*. The population of Proteobacteria had an average abundance of 1.40% *(SEM = 0.73%)* in the NC. This population ranged from <1% to 5% apart from the $BF_{OIL}$ stimulation, where the abundance increased to 6%. The population of Spirochaetota began the experiment at 3% and increased its abundance to 9% on day 11, at the end of the experiment. In the NC, the mean abundance was 8.13% *(SEM = 2.31%)*. Over the course of the experiment, the negative controls showed an increase solely in Spirochaetota between days 4 and 8, with an increase of 110.81% *(p = 0.025, δ = 1)*. The mean abundance of the methanogens in the NC was 2.55% *(SEM = 1.10%)*. Following an initial adaptation phase, the methanogens exhibited an abundance comparable to that observed in the pre-stimulation sample, except for the $BF_{OIL}$ stimulations. The methanogens or Euryarchaeota observed in the samples were mainly assigned to Methanomassiliicoccales, *Methanobrevibacter*, *Methanosphaera*, and Thermoplasmata. Their abundance was reduced with the progression of the incubation from 2.62% *(SEM = 0.65%)* to 0.83% *(SEM = 0.27%)* at day 8 and back to up to 1.72% *(SEM = 0.79%)* at day 11. The greatest reduction in Methanogens was observed in the $BF_{OIL}$ stimulations. The taxon of Bacteroidota did not differ significantly *(p > 0.025)* from the negative control in all samples. Fibrobacterota exhibited significant differences *(p = 0.025, Cliff's δ = −1)* in the 8.75 µM, 16.5 µM, and 33 µM 3-NOP, and the 1.25 µM $BF_{OIL}$ stimulations after 8 days, resulting in mean reductions of 76.45%, 65.74%, 67.45%, and 87.91%, respectively. Firmicutes were significantly increased *(p = 0.025, δ = 1)* in the 1.25 µM $BF_{OIL}$ stimulation on day 11, with an average of 39.08%. Methanogens showed a significant *(p = 0.025, δ = −1)* reduction in relative abundance in the 2.5 µM $BF_{OIL}$ set after 8 days, and in the 1.25 µM and 2.5 µM $BF_{OIL}$ groups after 11 days, with 80.03%, 97.66%, and 97.80% respectively. The Proteobacteria showed no significant *(p > 0.025)* differences compared to the negative control. Spirochaetota were significantly reduced *(p = 0.025, δ = −1)* for the 1.25 µM $BF_{OIL}$ stimulation on day 8 with 64.25%. Verrucomicrobia exhibited considerable decreases *(p = 0.025, δ = −1)* for 9 mg AT, for 1.25 µM, and 2.5 µM BF, and for 3.75 µM IF after 8 days, with reductions of 52.29%, 49.66%, 61.37%, and 57.37%, respectively. After 11 days, significant differences in the abundance of Verrucomicrobia were observed *(p = 0.025, δ = −1)* with 9 mg AT, 1.25 µM, 2.5 µM, and 3.75 µM BF, and IF, 8.75 µM, 16.5 µM, and 33 µM 3-NOP, and 1.25 µM and 2.5 µM $BF_{OIL}$, with reductions of 68.26%, 56.92%, 77.79%, 71.88%, 55.14%, 78.63%, 63.77%, 55.69%, 49.19%, 65.60%, 71.16%, and 78.80%, respectively. The abundances of the group "Other" were significantly reduced *(p = 0.025, δ = −1)* for the 1.25 µM and 2.5 µM $BF_{OIL}$ groups after 8 and 11 days with a reduction of 53.33%, 57.34%, 58.67%, and 56.98%, respectively. Unclassified counts were significantly elevated *(p = 0.025, δ = 1)* for 1.25 µM $BF_{OIL}$ after 8 and 11 days, with 129.07% and 49.94% respectively.

## Discussion

Despite active research on methanogenesis in the digestive system of ruminants due to its potential impact on global warming [16,55,56], the establishment of a robust, reliable, and affordable *in vitro* rumen cultivation system remains challenging in laboratories where expensive equipment is unavailable. However, such systems are necessary to investigate a multitude of rumen fermentation parameters over the course of time. With the development of our setup, we sought to combine the advantages of the two best-established systems, namely the throughput and ease-of-use of batch systems and the quality and stability of rumen simulation setups.

The apparatus was subjected to rigorous testing, and its application was demonstrated through a comparison of various anti-methanogenic additives. In the course of our evaluation of the system, we sought to utilize the most prominent and well-established anti-methanogenic compounds.

The results demonstrate the stability of the microbiome and the consistency of the overall gas production rates and the production rates of individual gas species. The experiments were conducted for a period of 11 days, with one instance extending to 23 days. Consequently, the readily replicable, self-constructed configuration is well-suited for rumen cultivation experiments that span days or even weeks.

The present study focused on three types of compounds. Namely, on AT as a natural product containing halogenated compounds, BF and the closely related IF as halomethane-based MCR inhibitors with potent, rapid effects, and 3-NOP as a nitro-ester MCR inhibitor with established but typically moderate suppression. $NC_{OIL}$ served as a matrix control, and to test $BF_{OIL}$ delivery, which may modulate release and stability [50]. The selection of concentration ranges was chosen to span sub-inhibitory to near-maximal effects reported *in vitro* to enable dose–response resolution [22,27,45,57].

A direct comparison of batch-culture incubations and RUSITEC setups has consistently revealed discrepancies in total gas and methane production. Our semi-continuous system yielded a total gas production rate of 156.21 mL/gDM/day *(SEM = 4.73 mL/gDM/day)*, with an average $CH_4$ fraction of 20.06% *(SEM = 0.84%)*. In a comparable batch flask system, Machado *et al.* incubated 125 mL of rumen culture with 1 g of Flinders grass and 0.2 g of decorticated cottonseed meal as a negative control for 72 hours [27]. Their experimental results yielded a total gas production of 126.8 mL/gOM *(SEM = 2.29 mL/gOM)* with a $CH_4$ output of 18.1 mL/gOM *(SEM = 0.61 mL/gOM)*, resulting in a CH4 fraction of 14.27%. Twenty-four hours after the start of the experiment, the total gas yield in their system was approximately 65 mL/gOM. On this basis, the daily gas output of our setup was approximately 156.50% higher, and the fractional $CH_4$ output was 40.57% higher, than that reported in Ref. [27], taking the reported conversion from OM to DM into account. The discrepancy is further emphasized by the 454.30% higher $CH_4$ output observed for our system, considering the augmented total gas production [27].

In contrast, the original RUSITEC system, as described in Ref. [40], yielded a total gas production of 72.2 mmol gas/day $(CH_4 + CO_2 + H_2)$ when supplied with 13.8 gDM/day for the incubation period of days 5–10 with a liquid turnover of 0.33 per day. This is equivalent to a total gas production of 117.27 mL/gDM/day with a $CH_4$ fraction of 25.18 mL/gDM/day under standard conditions. In comparison to these values, our experimental setup yielded a 33.21% higher total gas yield and 24.49% more $CH_4$ per unit of dry matter per day. Nevertheless, the proportion of $CH_4$ is notably higher for the RUSITEC, as evidenced by the 21.47% $CH_4$ contribution, which is 6.56% higher than the methane percentage observed in our experimental setup. Importantly, methane production in the RUSITEC system declined further during days 11–16, with $CH_4$ yield decreasing to 17.59%. This pattern likely reflects the progressive depletion of protozoa, which has been shown to reduce methanogenesis, and shift fermentation end products by diverting $H_2$ utilization. Such protozoal loss is a common feature of continuous-flow fermenters, contributing to their generally lower methane yields compared with short-term batch assays [39]. Another notable distinction of the RUSITEC configuration is its higher liquid turnover rate. As reported in Ref. [40], a low liquid turnover rate, such as the one employed in this study (0.2 per day), is generally associated with a reduced total output of the system. This observation is said to be attributed to the accumulation of metabolic end products, including volatile fatty acids and $H_2$, within the system [40].

Discrepancies in gas production across different studies may also be attributed to variations in inoculum sources, substrate compositions, and nutrient addition procedures. For example, slowly degradable feed may accumulate in the cultures, influencing fermentation dynamics. Additionally, our culture bottles contained ceramic cylinders, which facilitate bacterial attachment and retention of the solid fraction. The potential effects of these carriers, as well as the microbes attached to them, remain to be evaluated in future. Overall, the new system described here bears some resemblance to the RUSITEC setup, with the former likely exhibiting slightly elevated gas production.

For this study, inoculum was obtained from a randomized selection of rumen contents from animals exhibiting variation in terms of sex, breed, diet, health status, and potential previous exposure to antibiotic treatments. Consequently, the variability of the microbiome is pronounced and represents a cross-section of the cattle population raised in Bavaria, Germany. It has been demonstrated in previous studies that a variety of inoculum sources can still produce consistent and

reproducible fermentative measurements, particularly when experimental design incorporates sufficient replication and feeding standardization [58]. Consequently, the inocula from four distinct animals were pooled to mitigate the idiosyncrasies associated with inoculum [59]. Pooling rumen fluid from multiple donors per run aligns with established guidelines to mitigate donor-specific effects and enhance the external validity of screening studies [37].

The stability of the predominant phyla Bacteroidota and Firmicutes, along with the comparable abundance of Fibrobacterota, suggests that this setup exhibits properties similar to state-of-the-art cultivation systems. However, the high variance of the inocula and the different PCR primers complicate a direct comparison with published data [60,61]. This pattern is consistent with previous reports from RUSITEC and related continuous culture systems, where the core bacterial phyla remain stable across extended incubations [61,62]. In comparison with the ruminal microbiome sequences documented in Ref. [61], our proportion of Bacteroidota *(36.32%, SEM = 2.29%)* is consistent with their values (38.5%), while Firmicutes were lower in our system (35.61%, *SEM = 3.31%* vs. 56.4%), and Fibrobacterota (2.66%, *SEM = 1.49%*) remained within the range typically observed as a minor, yet consistent, fibrolytic phylum. The reduced proportion of Firmicutes in our fermenters is likely indicative of the carbohydrate-rich cultivation medium, which favored Bacteroidota specialized in starch and soluble sugar degradation, while limiting some Firmicutes that rely more on fiber fermentation [63].

All additives were tested on all rumen cultures, and the negative control cultures were derived from the same animals as the cultures utilized for stimulation experiments. Nonetheless, the initial rumen culture compositions exhibited variability across replicates of the experiments. Previous studies have shown a considerable variability in the reported amount of the initial fraction of Euryarchaeota [60,62,64]. During the incubation time, the overall abundance of methanogens fluctuated, yet the $CH_4$ production remained comparatively stable. This observation underscores the well-documented limitation of 16S rDNA metagenomic studies, which demonstrate that methanogen abundance does not necessarily correlate with functional methane output. The proportion of methanogens may fluctuate without corresponding alterations in activity because $CH_4$ flux is influenced by factors such as hydrogen availability, metabolic state, and interspecies interactions, rather than being solely dependent on archaeal counts alone. Recent studies have demonstrated a correlation between methane production and mcrA transcript levels in anaerobic digesters, rather than a correlation with archaeal 16S rDNA abundance [65]. Similarly, it has been shown that trophic interactions regulating $H_2$ supply substantially influence methanogenesis, resulting in disparities between archaeal abundance and $CH_4$ yield [66]. The notable exception in our study was stimulation with $BF_{OIL}$, where both $CH_4$ production and archaeal abundance underwent a substantial and irreversible decrease. In all other treatments, the disconnect emphasizes the need to integrate functional and activity-based analyses alongside 16S rDNA sequencing to fully understand the microbial drivers of methane production.

The demonstration experiments with our setup provide a valuable direct comparison of the potency of different $CH_4$ mitigation strategies, including the supplementation of 3-NOP, AT, BF, IF, and BF in an oil solution. The reductions in $CH_4$ production observed after the addition of AT and BF are consistent with the values reported in the literature [60]. Moreover, our findings indicated that 3-NOP consistently exhibited a weaker effect on methanogenesis than BF or IF. However, our results indicate a slightly better anti-methanogenic effect of 3-NOP than expected from literature values. Previously, a 76% reduction of $CH_4$ production has been reported for a supplementation of approximately 41.3 μM of 3-NOP [47]. In the present experiments, 33 μM of 3-NOP were administered once to achieve a maximum reduction of 74.63% (*SEM = 0.44%*) of $CH_4$ production. The administration of 3-NOP resulted in the most transient effect, with $CH_4$ production returning to its previous levels after a period of only five days.

In conclusion, the established methodologies and protocols, together with the conducted demonstration experiments, provide a robust methodological basis for future *in vitro* studies of methanogenesis. Such experimental approaches are essential for the development of anti-methanogenic additives. Moreover, the approaches can contribute to a better understanding of some of the controversial results from *in vivo* experiments in cattle [26,28,30,67,68], which suggest a need for further research in this area. The strengths of the developed system include long-term incubation, feedability, and daily sampling, which allow for more precise tracking of dynamic processes than short-term assays. Its main limitation is scalability, as flask size restricts large-scale

screening. Furthermore, once removed from the rumen, microbial communities inevitably shift due to altered nutrient supply and feeding intervals. These conditions, together with the absence of host factors and the accumulation of fermentation products, highlight the importance of interpreting in vitro results cautiously and validating findings *in vivo*.

## Supporting information

**S1 File. Step-by-step protocol collection;** *In vitro* **Cultivation and Microbiome Analysis, also available on proto-cols.io.**
(PDF)

**S2 File. Step-by-step protocol; Setup design, also available on protocols.io.**
(PDF)

**S3 File. Step-by-step protocol;** *In vitro* **cultivation, also available on protocols.io.**
(PDF)

**S4 File. Step-by-step protocol; Gas measurements, also available on protocols.io.**
(PDF)

**S5 File. Step-by-step protocol; Microbiome analysis, also available on protocols.io.**
(PDF)

**S6 File. Step-by-step protocol; Data evaluation, also available on protocols.io.**
(PDF)

**S1 Appendix. Additional Information.** Consisting of S1 Fig., S2 Fig., S1 Table, S2 Table.
(DOCX)

## Acknowledgments

We thank Volta Greentech AB (Götheborg, Sweden) for the kind gift of samples of the red algae, *Asparagopsis taxiformis*. We thank the staff members of Attenberger Fleisch GmbH & Co. KG Slaughterhouse in Munich for provision of samples and an introduction to probing the rumen. We are grateful to Dr. Markus Müller from the Institute for Chemical Epigenetics at LMU Munich for support of our compound synthesis. Some illustrative elements in Fig 2 were created by the authors with the assistance of ChatGPT-5.0 (OpenAI, San Francisco, USA) image generation.

## Author contributions

**Conceptualization:** Philip P. Laric, Armina Mortazavi, Benedikt Sabass.

**Data curation:** Philip P. Laric.

**Formal analysis:** Philip P. Laric.

**Funding acquisition:** Florian M. Trefz, Benedikt Sabass.

**Investigation:** Philip P. Laric, Kathrin Simon, Pauline S. Rittel.

**Methodology:** Philip P. Laric, Ewa Węgrzyn, Pauline S. Rittel.

**Project administration:** Florian M. Trefz, Benedikt Sabass.

**Resources:** Ewa Węgrzyn, Benedikt Sabass.

**Software:** Philip P. Laric.

**Supervision:** Florian M. Trefz, Benedikt Sabass.

**Validation:** Philip P. Laric, Armina Mortazavi.

**Visualization:** Philip P. Laric.

**Writing – original draft:** Philip P. Laric, Benedikt Sabass.

**Writing – review & editing:** Philip P. Laric, Armina Mortazavi, Pauline S. Rittel, Benedikt Sabass.

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
