## [Decision Letter · Decision Letter 0]

16 Jul 2025

PONE-D-25-30716A reliable *in vitro* rumen culture system and workflow for screening anti-methanogenic compoundsPLOS ONE

Dear Dr. Sabass,

Thank you for submitting your manuscript to PLOS ONE. After careful consideration, we feel that it has merit but does not fully meet PLOS ONE’s publication criteria as it currently stands. Therefore, we invite you to submit a revised version of the manuscript that addresses the points raised during the review process.

We look forward to receiving your revised manuscript.

Kind regards,

Bawadi Abdullah

Academic Editor

PLOS ONE

Journal Requirements:

“This Project was funded by the German Federal Ministry of Education and Research

(BMBF) under grant number 031B1504. PL and BS received funding

from the European Union’s Horizon 2020 research and innovation programme (grant

agreement No 852585).”

3. We note that you have a patent relating to material pertinent to this article. Please provide an amended statement of Competing Interests to declare this patent (with details including name and number), along with any other relevant declarations relating to employment, consultancy, patents, products in development or modified products etc. Please confirm that this does not alter your adherence to all PLOS ONE policies on sharing data and materials, as detailed online in our guide for authors http://journals.plos.org/plosone/s/competing-interests by including the following statement: "This does not alter our adherence to PLOS ONE policies on sharing data and materials.” If there are restrictions on sharing of data and/or materials, please state these. Please note that we cannot proceed with consideration of your article until this information has been declared.

4. We note that Figure 2 in your submission contain copyrighted images. All PLOS content is published under the Creative Commons Attribution License (CC BY 4.0), which means that the manuscript, images, and Supporting Information files will be freely available online, and any third party is permitted to access, download, copy, distribute, and use these materials in any way, even commercially, with proper attribution. For more information, see our copyright guidelines: http://journals.plos.org/plosone/s/licenses-and-copyright.

6. We note you have not yet provided a protocols.io PDF version of your protocol and/or a protocols.io DOI. When you submit your revision, please provide a PDF version of your protocol as generated by protocols.io (the file will have the protocols.io logo in the upper right corner of the first page) as a Supporting Information file. The filename should be S1_file.pdf, and you should enter “S1 File” into the Description field. Any additional protocols should be numbered S2, S3, and so on. Please also follow the instructions for Supporting Information captions [https://journals.plos.org/plosone/s/supporting-information#loc-captions]. The title in the caption should read: “Step-by-step protocol, also available on protocols.io.”

Please assign your protocol a protocols.io DOI, if you have not already done so, and include the following line in the Materials and Methods section of your manuscript: “The protocol described in this peer-reviewed article is published on protocols.io (https://dx.doi.org/10.17504/protocols.io.[...]) and is included for printing purposes as S1 File.” You should also supply the DOI in the Protocols.io DOI field of the submission form when you submit your revision.

If you have not yet uploaded your protocol to protocols.io, you are invited to use the platform’s protocol entry service [https://www.protocols.io/we-enter-protocols] for doing so, at no charge. Through this service, the team at protocols.io will enter your protocol for you and format it in a way that takes advantage of the platform’s features. When submitting your protocol to the protocol entry service please include the customer code PLOS2022 in the Note field and indicate that your protocol is associated with a PLOS ONE Lab Protocol Submission. You should also include the title and manuscript number of your PLOS ONE submission.

Additional Editor Comments:

Based on the reviewers’ evaluation, the current manuscript does not fully meet our stringent requirements for publication. However, we believe it holds potential and may be accepted pending major revisions. The required corrections are substantial and must address all concerns raised by the examiners. Authors are expected to revise the manuscript thoroughly and provide a detailed response to each comment. Only after careful consideration of the revised version will a final decision be made. We encourage the authors to review the feedback closely and make the necessary improvements to enhance the quality and clarity of their submission

Reviewers' comments:

Reviewer's Responses to Questions

**Comments to the Author**

1. Does the manuscript report a protocol which is of utility to the research community and adds value to the published literature?

Reviewer #1: Yes

Reviewer #2: Yes

2. Has the protocol been described in sufficient detail?

To answer this question, please click the link to protocols.io in the Materials and Methods section of the manuscript (if a link has been provided) or consult the step-by-step protocol in the Supporting Information files.

The step-by-step protocol should contain sufficient detail for another researcher to be able to reproduce all experiments and analyses.

Reviewer #1: Partly

Reviewer #2: Yes

3. Does the protocol describe a validated method?

Reviewer #1: Yes

Reviewer #2: Yes

4. If the manuscript contains new data, have the authors made this data fully available?

Reviewer #1: Yes

Reviewer #2: N/A

**5. Is the article presented in an intelligible fashion and written in standard English?**

Reviewer #1: Yes

Reviewer #2: Yes

6. Review Comments to the Author

Reviewer #1: Overall, the manuscript is comprehensive and addresses an important issue. However, several aspects should be improved. Major revisions are suggested below to enhance the manuscript:

• The abstract is too general. Include specific results.

• Provide more quantitative data (e.g., effect sizes, statistical comparisons) to support claims about system performance.

• What is this “everyday labware”. It is vague and informal for a research article.

• There are formatting mistakes throughout the article. Like font style, alignment, and subscript. The figure numbers should be checked again.

• All the figures are not clear. Reproduce them.

• Check the reference style

• Please provide the list of abbreviations or table for abbreviations

• The discussion often lacks clarity in structure and logical flow, making it difficult to follow key conclusions.

• Discussion should include statistical comparisons or confidence intervals to assess significance across systems.

• The discussion lacks targeted reasoning.

• Enhance the interpretation of microbiome data, particularly the discrepancy between methanogen abundance and CH4 output.

• The authors should justify use of highly variable inocula.

• The final paragraph lacks a critical reflection on the system’s limitations, such as microbial drift over time, absence of host factors, or scalability to large screening efforts.

• The discussion is too short. It must be revised and explain every aspect in detail.

Reviewer #2: The authors tried to present an overview of anti-methanogenic compounds to mitigate the emission of these GHGs. However, there are several parts that need to be revised before publication.

1. The abstract of the article needs to be revised as it lacks key findings in the article.

2. The author should clarify why they chose T powder, BF, IF, 3-NOP, and rapeseed oil (NCOIL), and explain the rationale behind the concentrations of these supplements.

3. The image quality in the paper is poor. The author should revise these figures to improve readability.

4. The author should mention the material along with the method.

5. There is a lot of referencing to supplementary files (for example, S1 Fig, S2 File), but the main text lacks context.

6. There are various typo errors, such as "abbundances" (line 325). The author must improve and avoid these errors.

7. PLOS authors have the option to publish the peer review history of their article (what does this mean? ). If published, this will include your full peer review and any attached files.

**Do you want your identity to be public for this peer review?** For information about this choice, including consent withdrawal, please see our Privacy Policy .

Reviewer #1: **Yes: ** Syed Muhammad wajahat ul hasnain

Reviewer #2: No

---

## [Author Response · Author response to Decision Letter 1]

30 Sep 2025

Dear Dr. Syed Muhammad wajahat ul hasnain, dear Rewiver #2,

We would like to sincerely thank both reviewers for their thoughtful and constructive comments. We greatly value the time and expertise you invested in evaluating our work. Your feedback has been valuable in enhancing the clarity, rigor, and overall quality of our manuscript. Also in "Response to Reviewers".

We respond below in the order presented by the editor.

Reviewer #1

1. The abstract was revised to include key quantitative results, specifically the maximum CH₄ reductions for each additive, with supporting statistical values.

2. The Results and Discussion sections were expanded to include detailed cross-system comparisons, effect sizes, and standard errors where available.

3. This phrase was replaced with a more precise description of the equipment used.

4. All formatting, subscripts, figure numbers, units, and taxonomy were standardized.

5. All figures were re-exported at higher resolution and checked with PACE for compliance, with improved clarity, and consistent formatting. Figure 2 was redrawn using CC BY 4.0–compliant artwork.

6. References were checked throughout and corrected for consistency and accuracy.

7. A comprehensive list of abbreviations was added before the references

8. The Discussion section was substantially restructured and expanded. More paragraphs were added for clarity, targeted reasoning was included, and statistical comparisons and microbiome interpretation were strengthened. Inoculum variability was justified, and a detailed limitations section was incorporated.

Reviewer #2

1. The abstract was revised to include key results, including maximum CH₄ reductions.

2. A rationale paragraph was added to the Discussion, explaining the choice of compounds (AT, BF, IF, 3-NOP, NCOIL) and the basis for the selected concentration ranges.

3. Figures were replaced with higher-resolution versions and checked with PACE for compliance. Fig 2 was redrawn using CC BY 4.0–compliant artwork.

4. Materials are detailed in the deposited protocols on protocols.io, which are referenced in the Materials & Methods section and attached as S1–S6 Files with captions.

5. In the main manuscript there are no references to appended files or figures beyond the supporting information. As this is a protocol paper, the main text is not designed to be all-encompassing; the detailed step-by-step procedures are provided on protocols.io and as S1–S6 Files in line with PLOS ONE requirements.

6. The manuscript was thoroughly revised for language and formatting, with all typographical and taxonomic errors corrected.

We hope these revisions fully address the journal’s requirements. We look forward to your assessment.

Sincerely Philip Laric, on behalf of all authors

---

## [Decision Letter · Decision Letter 1]

16 Oct 2025

A reliable *in vitro* rumen culture system and workflow for screening anti-methanogenic compounds

PONE-D-25-30716R1

Dear Dr. Sabass,

We’re pleased to inform you that your manuscript has been judged scientifically suitable for publication and will be formally accepted for publication once it meets all outstanding technical requirements.

Kind regards,

Bawadi Abdullah

Academic Editor

PLOS ONE

Additional Editor Comments (optional):

recommended based on the review comments.

Reviewers' comments:

Reviewer's Responses to Questions

**Comments to the Author**

1. Does the manuscript report a protocol which is of utility to the research community and adds value to the published literature?

Reviewer #1: Yes

Reviewer #2: Yes

2. Has the protocol been described in sufficient detail?

To answer this question, please click the link to protocols.io in the Materials and Methods section of the manuscript (if a link has been provided) or consult the step-by-step protocol in the Supporting Information files.

The step-by-step protocol should contain sufficient detail for another researcher to be able to reproduce all experiments and analyses.

Reviewer #1: Yes

Reviewer #2: Yes

3. Does the protocol describe a validated method?

Reviewer #1: Yes

Reviewer #2: Yes

4. If the manuscript contains new data, have the authors made this data fully available?

Reviewer #1: N/A

Reviewer #2: No

**5. Is the article presented in an intelligible fashion and written in standard English?**

Reviewer #1: Yes

Reviewer #2: Yes

6. Review Comments to the Author

Reviewer #1: The revised manuscript has addressed the earlier concerns, and the authors have made significant improvements to strengthen the overall quality of the paper. The methodology is clearly presented, and the results are now better supported with relevant discussion. I recommend the paper for acceptance in its present form without any further changes.

Reviewer #2: The manuscript has officially been accepted in its current form, and no additional changes or corrections are necessary before publication.

7. PLOS authors have the option to publish the peer review history of their article (what does this mean? ). If published, this will include your full peer review and any attached files.

**Do you want your identity to be public for this peer review?** For information about this choice, including consent withdrawal, please see our Privacy Policy .

Reviewer #1: No

Reviewer #2: No

---

## [Editor Report · Acceptance letter]

PONE-D-25-30716R1

PLOS ONE

Dear Dr. Sabass,

I'm pleased to inform you that your manuscript has been deemed suitable for publication in PLOS ONE. Congratulations! Your manuscript is now being handed over to our production team.

Kind regards,

on behalf of

Dr. Bawadi Abdullah

Academic Editor

PLOS ONE